# Heavy Metals in Sediments of Hulun Lake in Inner Mongolia: Spatial-Temporal Distributions, Contamination Assessment and Source Apportionment

Tong Liu [1,†], Dasheng Zhang [1,†], Weifeng Yue [2,*], Boxin Wang [1], Litao Huo [1], Kuo Liu [1] and Bo-Tao Zhang [2,*]

1   Hebei Institute of Water Science, Shijiazhuang 050051, China; tongliu2020@163.com (T.L.)
2   College of Water Sciences, Beijing Normal University, Beijing 100875, China
*   Correspondence: yuewf@bnu.edu.cn (W.Y.); zhangbotao@126.com (B.-T.Z.)
†   These authors contributed equally to this work.

**Abstract:** The spatial and temporal distributions, contamination evaluation, and source apportionment of Cu, Zn, As, Pb, Cd, and Cr in the sediments of Hulun Lake were explored in this work. The pollution characteristics of six heavy metals were assessed by single factor pollution index (PI) and geo-accumulation index (Igeo). The sources of heavy metals in the surface sediments were analyzed by the positive definite matrix factorization (PMF) and Pearson correlation analysis. The sedimentary records of heavy metals in core sediments were reproduced by radioisotopes. The average concentrations of 6 heavy metals except Cd were lower than the corresponding background values. The spatial distributions of Cu, Zn, Cr, Cd and As were generally similar and showed higher abundances in the southwestern part of the lake. With the use and import of heavy metals, the concentration of heavy metals in core sediments increased with the fluctuation of years. The peak of heavy metal concentration was related to the high growth rate of gross domestic product in 2003-2008. The single factor pollution index and geo accumulation index results showed that the surface sediment was mainly polluted by Cd, followed by Zn and As. Natural parent material, agricultural activities and industrial activities were the main sources of heavy metal pollution in the sediments, accounting for 17.03%, 26.34%, and 56.63% of the total heavy metal accumulation, respectively. Pb was derived mainly from natural parent material. Cd and As were closely associated with agricultural activities. Cu and Zn were mainly attributed to industrial mining activities. Source apportionment of the ecological risks of heavy metals illustrated that industrial sources were the primary ecosystem risk sources (66.1%), followed by agricultural sources (23.75%) and natural sources (10.15%). The results will also provide reference data for future studies of heavy metals pollution in sediments from Hulun Lake and other lakes.

**Keywords:** heavy metal; sediment; spatiotemporal distribution; source apportionment; risk assessment Hulun Lake



## 1. Introduction

With the rapid development of industrialization and urbanization in recent decades, the increase of metal pollution in aquatic ecosystems has become one of the critical environmental problems that attract extensive attention because of their toxicity, bioavailability and mobility [1,2]. The hazards of heavy metals in the environment are not only related to their total concentration, but are also attributed to chemical partitioning. The sources of heavy metals in the aqueous environment are broadly classified as natural or anthropogenic. Weathering of bedrock and volcanic eruptions are categorized as natural sources. Anthropogenic sources include metal ore and coal mining, production and domestic wastewater, fertilization, fossil fuel combustion, tourism, aquaculture and fisheries, etc. [3–5]. The primary sources of heavy metals vary on a continental scale. In Africa, heavy metals mainly come from bedrock weathering, metallurgy, industry, and domestic wastewater [6,7]. The

density of heavy metals (such as Cu, Pb, Zn, Cr, As and Cd) usually exceeds 4.5 g/cm$^3$ [7]. Some heavy metals (e.g., Pb and Cd) are mainly classified as toxic substances with serious effects on the aquatic environment and human health. Although V, Cr, Co, Ni and Zn are essential for specific biological processes, they are defined as toxic at higher concentrations [8,9]. Enrichment of heavy metals in lakes causes fatal harm to human health and aquatic organisms. Therefore, it is necessary to place a high premium on the heavy metals pollution in lakes. Heavy metal pollution control in lakes should receive more and more attention [10].

The prevention and control of heavy metal pollution in lake sediments is one of the best strategies due to the difficult and time-consuming degradation of heavy metals from the sediments [11]. To present, many studies have been carried out on the potential risk assessment and sources of heavy metals in lake sediments [12]. To quantify the contamination and ecological risk of metals in sediments, several indices have been proposed, which can be summarized as the geoaccumulation index (Igeo), the pollution index (PI) and the potential ecological risk Index (PERI) methods. Analyzing the distribution characteristics, PERI and potential sources of heavy metals in lake sediments is a necessary prerequisite for taking effective pollution control measures. The source of heavy metals is varied, such as machinery manufacturing, pesticides, fertilizers, exhaust emissions, mining, and natural parent materials. Hence, more research should be devoted to quantifying the source of heavy metals and preventing them from entering lakes [12]. The positive matrix factorization (PMF) is a quantitative method for pollutant source matrix analysis based on a receptor model that is not limited by the composition of a single pollutant source. In addition, it has been extensively used in the identification and quantification of heavy metals pollution sources [13].

Hulun Lake is located in the western part of Hulunbeier Grassland, known as the lung of the grassland, which brings together the rivers of Kelulun, Wuerxun, and Xinkai [14]. In recent years, with the economic and social development around the lake and the impact of global warming, the lake surface has been shrinking, grassland degradation and ecological environment and the water quality is deteriorating, fishery resources are on the verge of exhaustion, and the wetland ecological environment is in urgent need of protection [15]. At present, research on heavy metals in Hulun Lake mainly focuses on content distribution and pollution assessment, while previous studies focused on source analysis and vertical distribution characteristics.

To better protect fishery resources and environment in Hulun Lake waters and maintain aquatic environment, the objectives of this research are (1) to illustrate heavy metals content in the surface sediments of Hulun Lake, (2) to analyze the spatial distribution of heavy metals in surface sediments, (3) to assess heavy metals pollution in surface sediments, (4) to identify and analyse the potential sources of heavy metals and the contributions of the sources to ecological risk by the PMF model, and (5) to disclose heavy metals temporal variation in the core sediments. The overall objective of this study was to assess the spatial and temporal distribution of heavy metals in sediments, contamination evaluation, and the preferential contribution to ecological risk and source allocation. The results should provide basic data for the health of Hulun Lake aquatic ecosystem.

## 2. Materials and Methods

### 2.1. Study Area

Hulun Lake basin (including Hailar River basin and Halaha River basin) is located in China and Mongolia, with a drainage area of $2.92 \times 10^5$ km$^2$. With an area of 2339 square kilometers and an average water depth of 5.6 m, Hulun Lake is the fourth largest freshwater lake in China [16]. The lake water is mainly supplied by precipitation, surface runoff, and groundwater. The runoff of the Wuerxun River and the Kelulun River is the primary water source of Hulun Lake, and the Xinkai River is the outlet river. The Hailar basin around the lake is active in magmatic activity during the geological period and contains rich metal mineral resources such as Cu, Zn, Pb, Mn, Ag, Fe, etc.

### 2.2. Sample Collection and Physicochemical Analysis

In this study, Figure 1 shows the sample site locations of Hulun Lake in Inner Mongolia. The sediment samples from Hulun Lake (n = 12) were collected in August 2020. The surface sediment samples were gathered at a depth of 2 cm by a stainless steel sampler (JDHC-200A, Jintan, Changzhou, China). The core sediment sample C1 was gathered from the central part of the lake, cut sequentially at 2 cm thickness. Each sample was then transferred to a clean polyethylene sample bag and labeled, transported as quickly as possible back to the lab, and sand, gravel, and plant roots were removed. Stored at 4 °C until further analysis.

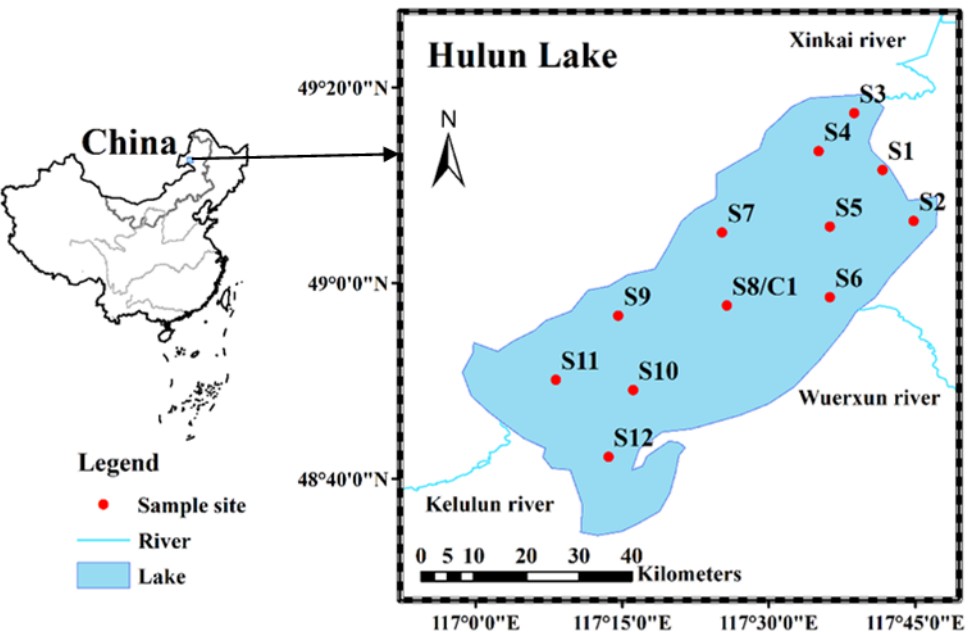

**Figure 1.** Locations of the sampling sites in Hulun Lake.

To detect heavy metals concentrations, each 8 g of sample was weighed with an electronic scale (XPR204S/AC) and dried in an oven (GN-53A) at 100 °C for 12 h, and then ground and sieved through a 100-mesh sieve (GILSON ISO 565). Eventually, Heavy metal content is measured using X-ray fluorescence spectrometry (Hitachi, Shanghai, China) and each sample was repeated three times. The core sediments' [210]Pb and [137]Cs specific activities were measured by a GWL high-purity germanium gamma spectrometer (AMETEK-AMT Ortec Co., Santiago, MN, USA). The China Institute of Atomic Energy supplied standard isotopic samples. Sediment age series calculated by constant recharge rate model.

### 2.3. Assessment Method

#### 2.3.1. Single Factor Pollution Index (PI)

The single factor pollution index (PI) method refers to the method for assessing the pollution degree of a certain heavy metals in soil or sediment [17], as shown in Equation (1):

$$PI = CI/SI \tag{1}$$

where *CI* and *SI* mean actual content and the background value of heavy metals i, respectively.

#### 2.3.2. Geo-Accumulation Index (Igeo)

The Igeo was originally proposed by Muller for application to bottom sediments [18]. This method has also been applied to identify anthropogenic sources of heavy metals in sediments [19]. The Igeo of heavy metals was calculated according to the follows:

$$Igeo = \log_2[C_n/(1.5B_n)] \tag{2}$$

where *Cn* is the concentration of heavy metals measured in sediment and *Bn* is their baseline shown in Table 1. 1.5 is the background matrix correction coefficient. The geo-accumulation index consists of 7 grades or classes as shown in Table S1 [20].

**Table 1.** Statistical results of heavy metals in Hulun Lake sediments.

| Category | Heavy Metals (mg/kg, n = 12) | | | | | |
|---|---|---|---|---|---|---|
| | **Cu** | **Pb** | **Cr** | **Zn** | **Cd** | **As** |
| Maximum | 30.21 | 28.93 | 34.11 | 116.10 | 0.85 | 14.8 |
| Minimum | 2.96 | 5.39 | 9.20 | 24.50 | ND | 1.50 |
| Mean | 19.32 | 17.63 | 21.90 | 71.36 | 0.33 | 7.22 |
| S.D. | 10.77 | 8.38 | 9.21 | 31.56 | 0.25 | 4.8 |
| CV(%) | 55 | 47 | 41 | 43 | 75 | 66 |
| LEL | 16 | 31 | 26 | 120 | 0.6 | 6 |
| SEL | 110 | 250 | 110 | 820 | 10 | 33 |
| Background value [a] | 35 | 35 | 90 | 100 | 0.2 | 12 |

S.D.: Standard deviation; CV: Coefficients of variation; ND: Not detected; LEL: Lowest effect level; SEL: Severe effect level [21]; [a] Soil background concentrations of heavy metals in Hulun Lake basin [15].

*2.4. Receptor Model PMF 5.0*

The PMF model was used as the source analysis method of heavy metals in sediments [22,23]. The calculation method is shown in Equation (3):

$$X_{ij} = \sum_{k=1}^{p} g_{ik} f_{kj} + e_{ij} \tag{3}$$

where $X_{ij}$ means content of the $j_{th}$ heavy metals at the $i_{th}$ sampling site, $g_{ik}$ represents the contribution of the $k_{th}$ source to the $i_{th}$ sample, $f_{kj}$ means the content of the element $j$ from the $k_{th}$ source, and $e_{ij}$ means the residual error matrix. The objective function $Q$ is obtained in Equation (4):

$$Q = \sum_{i=1}^{n} \sum_{j=1}^{m} \left( \frac{e_{ij}}{u_{ij}} \right)^2 \tag{4}$$

where $u_{ij}$ denotes the uncertainty of the $j_{th}$ heavy metal in the $i_{th}$ sample, calculated from the species-specific method detection limit (MDL), the concentration and the error component provided [24]. If the heavy metals concentration exceeds the MDL, the $u_{ij}$ is obtained as described in Equation (5) [25]:

$$u_{ij} = \sqrt{(error fraction \times concentrations)^2 + (MDL)^2} \tag{5}$$

*2.5. Statistical Analysis*

The spatial distribution pattern of PAEs in the sediments was mapped by kriging spatial interpolation analysis using ArcGIS 10.6 software. Data processing was completed by Origin Pro 9.0. Pearson correlation analysis with IBM SPSS Statistics 25 was used to investigate the relationship between metal content.

### 3. Results and Discussions

*3.1. Heavy Metals Content in the Surface Sediments of Hulun Lake*

Table 1 described the concentration of 6 heavy metals in the surface sediments of Hulun Lake. The ranges of Cu, Pb, Cr, Zn, Cd, and As content were 2.96–30.21 mg/kg, 5.39–28.93 mg/kg, 9.20–34.11 mg/kg, 24.50–116.10 mg/kg, ND-0.85 mg/kg and 1.50–14.8 mg/kg, respectively. Except for Cd, the average concentrations of these heavy metals were below the corresponding background values. The maximum concentrations of Zn and As were higher than the background concentrations. The coefficient of variation (CV) can visually indicate the changes in the regional distribution of heavy metals. The higher CV value, the stronger the disturbance of anthropogenic activities on the regional dis-

tribution of heavy metals. The CV of Cu, Cd, and As were all greater than 50%, indicating that the regional distribution of these elements was highly different in sediments.

The SQG of the Canadian Council of Ministers of the Environment (CCME) in Ontario classifies the impact of heavy metals on sediments as No Effect Level (NEL), Lowest Effect Level (LEL), and Severe Effect Level (SEL) [21]. This research has concluded that the LEL and SEL have the highest quantities of Cu, Cd, Cr, and As. In comparison to LEL, Pb and Zn have lower maximum concentrations.

To compare the pollution level of Hulun Lake, mean concentration of heavy metals in sediment of different lake systems are summarized in Table 2 following the order of sampling time. The concentrations of Cu, Pb, Cr, Zn, Cd, and As are lower than in lakes such as Dongjiang Lake, Hongze Lake, Taihu Lake, and Poyang Lake, which were probably related to population density and anthropogenic activities intensity. Taihu Lake has the highest concentration of heavy metals compared with other domestic lakes. The Taihu Lake basin is one of the regions with the most concentrated population and the fastest urbanization process in China. This region's rapid economic and social development has caused increasingly major water pollution problems. Although the catchment area of Hulun Lake is 256,000 km$^2$, the basin is remote and sparsely populated, with over 60% of the Hulun Lake basin located in Mongolia. The population density in China's watershed areas is close to ~1.6/km$^2$, which is much lower than other lake basins.

**Table 2.** Mean concentration of heavy metals in sediments of the lake systems following the order of sampling time.

| Lake | Sampling Year | Heavy Metals (mg/kg) | | | | | | References |
|---|---|---|---|---|---|---|---|---|
| | | Cu | Pb | Cr | Zn | Cd | As | |
| Dongjiang Lake, China | 2021 | 33.01 | 47.40 | 67.58 | 113.9 | 2.25 | 80.80 | [26] |
| Kangryong River, North Korea | 2019 | 32.70 | 30.90 | 53.90 | 132.50 | 0.28 | 14.30 | [27] |
| Hongze Lake, China | 2018 | 25.35 | 27.2 | 66.78 | 74.77 | 0.23 | 16.55 | [28] |
| East Dongting Lake, China | 2017 | 42.60 | 58.60 | 42.9 | 88.80 | 5.50 | ND | [29] |
| Taihu Lake, China | 2017 | 44.71 | 37.00 | 102.46 | 163.62 | 0.80 | 13.34 | [30] |
| Persian Gulf, Iran | 2017 | 14.87 | 7.42 | 59.22 | 32.51 | 0.15 | 5.92 | [31] |
| Lake Emerald, India | 2017 | 611.32 | 34.04 | 411.48 | 174.40 | ND | ND | [32] |
| Inle Lake, Myanmar | 2017 | 7.92 | 9.64 | 19.24 | 18.80 | 0.05 | 6.59 | [2] |
| Los Molinos Lake, Argentina | 2015 | 4.67 | 6.45 | 2.50 | 9.80 | 0.06 | 1.74 | [33] |
| Poyang Lake, China | 2014 | 35.17 | 32.63 | 81.39 | 104.17 | 0.66 | 11.34 | [34] |
| The present study | 2020 | 19.32 | 17.63 | 21.90 | 71.36 | 0.33 | 7.22 | - |

*3.2. Spatial Distribution of Heavy Metals in Surface Sediments*

The continuous spatial distribution profiles of heavy metals in the surface sediments were analyzed using ArcGIS to visualize the concentration level of heavy metals between different sampling sites, as shown in Figure 2. The distribution characteristics of heavy metal contents in the surface sediments of the study area are different. The concentration of Pb is highest in the Wuerxun estuary. Cu, Zn, Cr, Cd, and As exhibited roughly similar spatial distributions, showing higher abundance in the southwest of the lake district. There was a tourist resort and No. 5 fishing ground southwest of Hulun Lake, and the frequent anthropogenic activities might be the main cause. There were more than 70 oil wells around Wuerxun River, and the higher Pb concentration might originate from exploitation [35,36]. The results show that pollution sources and artificial activities impact the spatial distribution of heavy metals in the catchment area of specific lakes. As a result, most of the heavy metals were likely to be contaminants of terrestrial origin, and surface runoff and adsorption to suspended particulate matter might be the pathways that carry contaminants from the catchment to the lake.

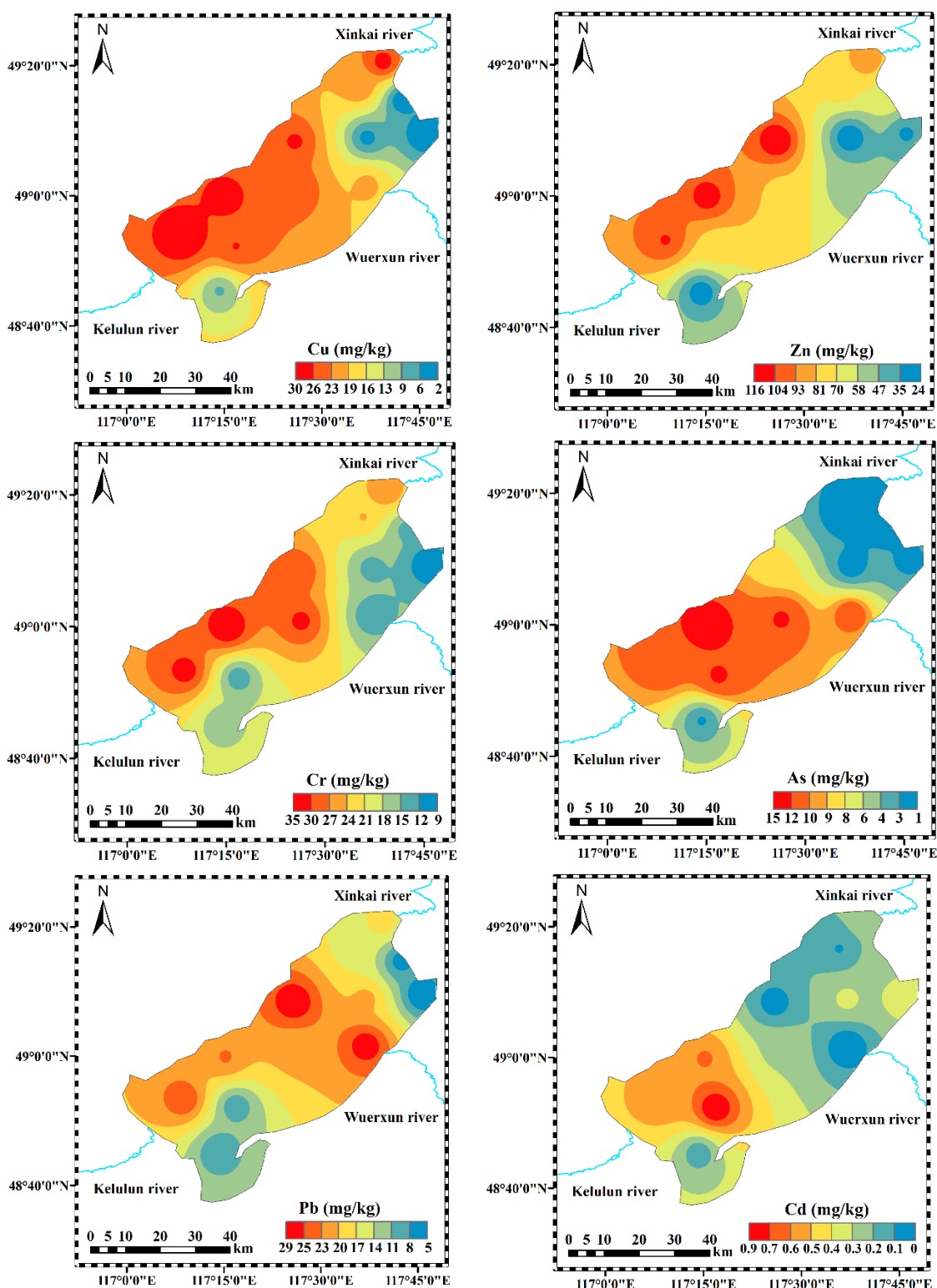

**Figure 2.** Spatial distribution patterns of heavy metals in the surface sediments of the Hulun Lake.

The CV of heavy metals pollution is a statistic reflecting the distribution uniformity and varying degree of heavy metals elements in the study area. The greater the CV value, the greater the disruptions of anthropogenic factors on its content distribution and the greater the difference in its spatial distribution [37]. Under the value of CV, the heavy metals variability in sediments could be divided into low variation (≤15%), medium variation (15–35%), and high variation (>35%) [38]. The CV values of all heavy metals in the surface

sediments of the study area are Cd > As > Cu > Pb> Zn >Cr in descending order in Table 1. All heavy metals show high variation characteristics, indicating that the spatial distribution and sources of heavy metals are different and disturbed by anthropogenic activities.

### 3.3. Assessment of Heavy Metals Pollution in Surface Sediments

Single factor pollution index (PI) refers to the method of evaluating the pollution grade of a certain heavy metals in soil or sediment [39]. The PI results for Pb, Cr, Cu, As, Cd, and Zn are shown in Figure 3. The PI. varied considerably across the different metals. In this work, the PI values were classified as low (PI < 1), moderate (PI between 1 and 3), high (3 < PI ≤ 5), and extremely high (PI > 5). The PI values for Cd in sediments varied from 0 to 4.25, with about 16.7% of the analyzed samples presenting high PI for Cd. Zn and As present moderate contamination, with PI values between 0.25 and 1.1 for Zn and between 0.13 and 1.2 for As. Moderate PI values were found in 25% of the samples for Zn and 42% for As. These results indicated that Pb, As, and Zn pollution might be widespread in Hulun Lake. Cu, Pb, and Cr exhibited low pollution, with PI values below 1.

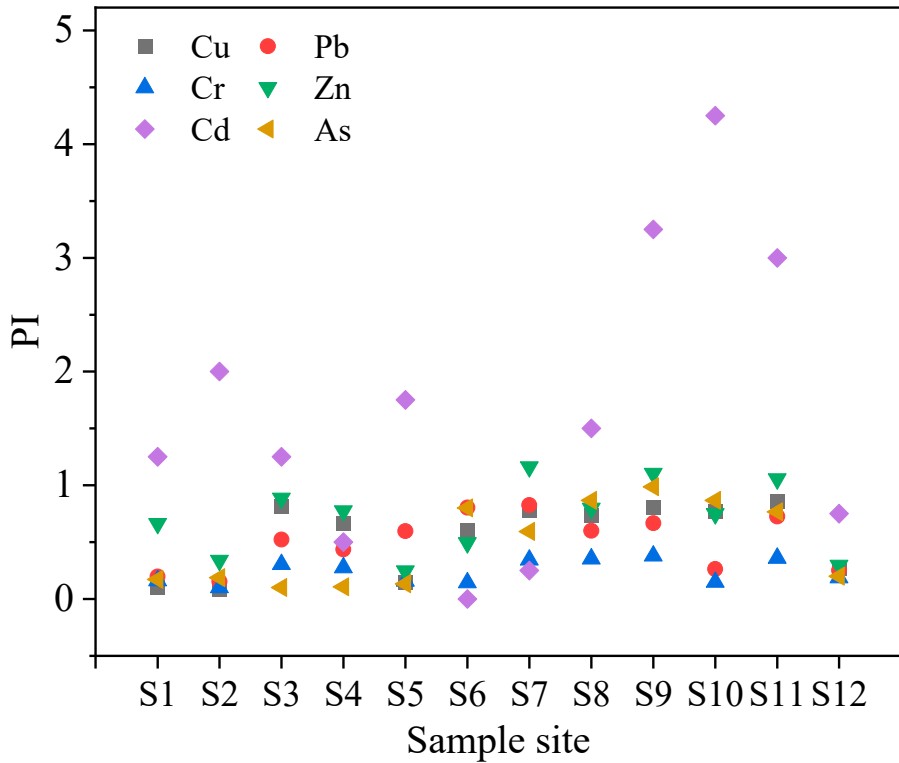

**Figure 3.** Single factor index (PI) distribution of heavy metals in surface sediments of Hulun Lake.

The Igeo is a widespread tool to assess the contamination degree of sediment. The essence is to remove the corresponding natural or background content from the present heavy metals content to obtain the total enrichment degree of heavy metals caused by anthropogenic activities. The Igeo variation characteristics of heavy metals in the sediments were presented in Figure 4. In the sediments of Hulun Lake, the Igeo of Cd was the highest, with 16.7% at the moderate pollution level and 25% between unpolluted to moderately pollution level. The reason might be related to the discarded batteries of seine netting fishing [40]. In addition, Zn reached a moderate pollution level of 8.3%. Cu, Pb and Cr showed unpollution level of 100%, indicating that these three heavy metals do not significantly contaminate the sediments. The results of the Igeo and PI are basically consistent. The study of heavy metal pollution in Hulun Lake had important research value. In the evaluation of heavy metal pollution, a combination of multiple evaluation methods was used to make the evaluation results more accurate and reference value.

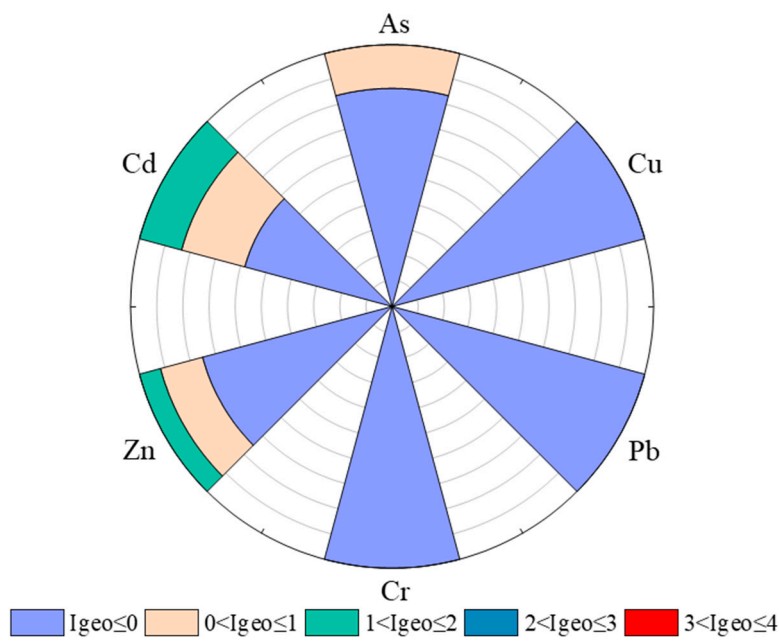

**Figure 4.** Geo-accumulation rose chart of respective heavy metals in the sediment of Hulun Lake.

*3.4. Source Identification of Heavy Metals in Surface Sediments by PMF*

Correlation and PMF analyses have been extensively applied to study the sources of heavy metals [41]. The correlation between heavy metals can qualitatively explain their sources and transport [42]. The correlation coefficients of heavy metals concentrations was exhibited in Figure 5. There were stronger correlations among Zn, Cr, and Cu. Cd showed a significant positive correlation with As. Pb exhibited a positive correlation with Cr and Cu. This result reflects the homology of heavy metals in sediments and is consistent with the spatial distribution characteristics of heavy metals mentioned above. The highly correlated results for these heavy metals suggested that they might belong to the same sources with similar levels of contamination and release [43].

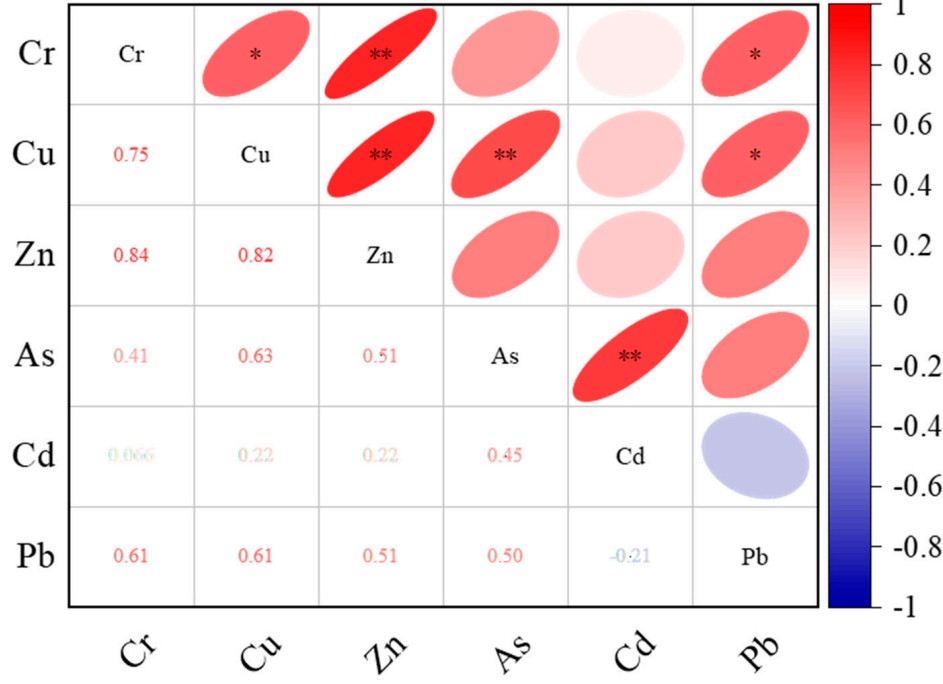

**Figure 5.** Pearson correlation coefficients among heavy metals (** $p < 0.01$, * $p < 0.05$).

The PMF model was intensively applied to the source identification and concentration quantification of heavy metals in sediments. EPA PMF 5.0 software was used to examine the amounts and unknown concentrations of six heavy metals in surface sediment samples. The analysis results are shown in Figures 6 and 7a. Factor 1 contributed 17.03% of the heavy metals sources in the sediment, among which Pb (52.10%) has a higher loading. The maximum concentration of Pb was below the local background value, reflecting Pb might not be externally contaminated. In the pollution assessment of PI and Igeo, Pb showed a pollution-free level and could be considered to come from the natural background. Therefore, factor 1 was defined as the natural parent material source.

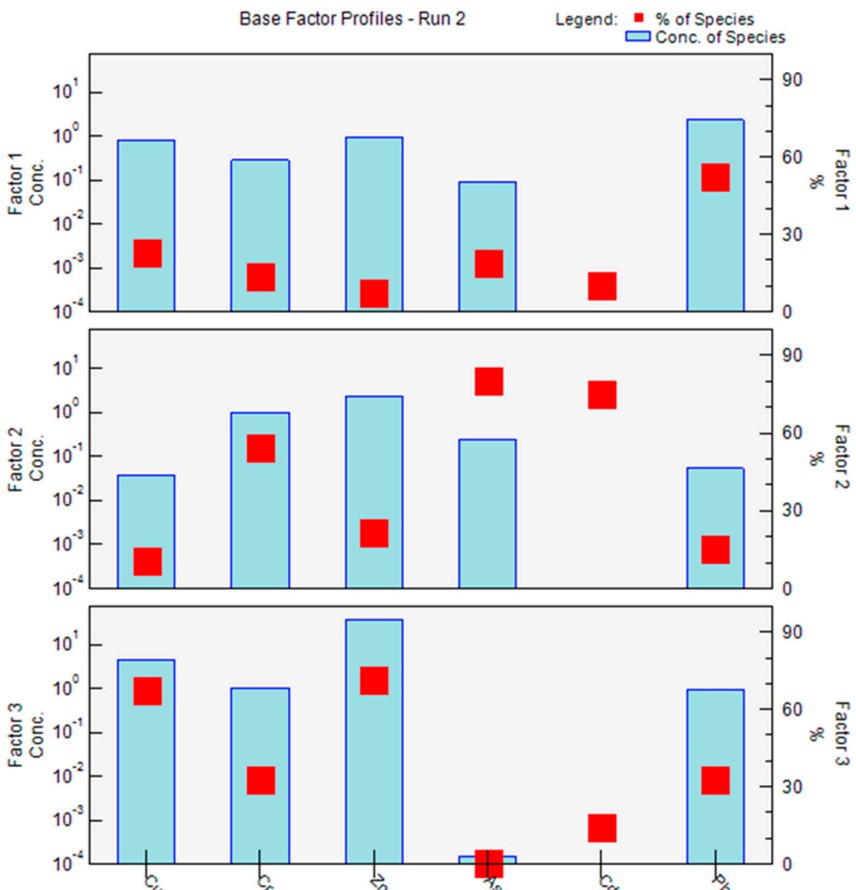

**Figure 6.** Source profiles and contributions of heavy metals in the studied Hulun Lake sediments obtained with the PMF model.

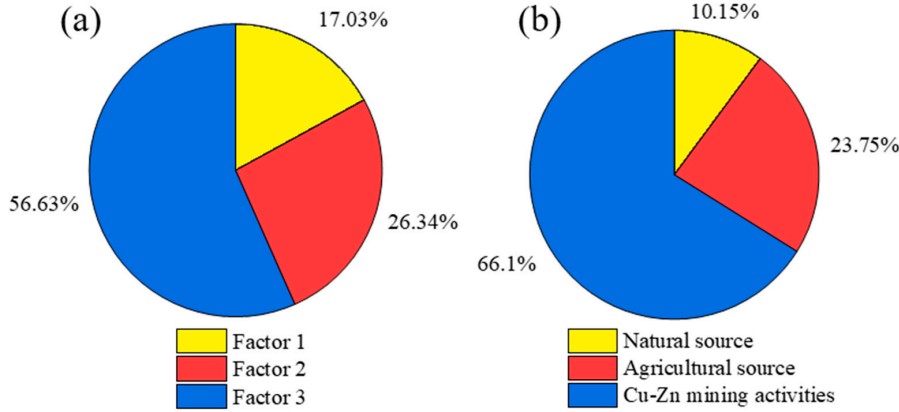

**Figure 7.** Contributions of various sources to the sediment heavy metal concentrations (**a**) and the associated ecological risks (**b**) in the study area.

Factor 2 had major factor loadings for As, Cu, and Cd and accounted 26.34% of the contributions of various heavy metals sources to the heavy metals concentrations in the sediments of Hulun Lake. Sewage agricultural irrigation and some agricultural products such as phosphate fertilizers, pesticides and organic fertilizers emitted large amounts of Cd [44,45]. According to researchs, the accumulation of Cr, As, and Cd was closely related to agricultural production activities, e.g., long-term application of pesticides [45], especially the content of Cd in sediment is significantly related to the content of total phosphorus. Animal husbandry is developed around the Kelulun River and Wuerxun River. Meanwhile, the water source of Hulun Lake comes from the runoff input of the Urxun River and Kulun River, and the final pollutants flow into the lake through the surface runoff and accumulate in the sediment. Therefore, factor 2 was associated with agricultural sources.

Factor 3 exhibited significant factor loadings for Zn and Cu and accounted a relatively large amount (56.63%) of the contributions of various sources of heavy metals to the heavy metals concentrations in the sediments of Hulun Lake. Cu and Zn are indicators of the source of mineral dust, which is closely related to human activities. There are many zinc copper mines in the north of Hulun Lake [46]. Mining activities result in the release of metals [47]. Some processing wastewater and metal dust will be discharged during metal smelting and mechanical processing. Correlation study results reveal high positive correlations between Cu and Zn and similar spatial distributions, demonstrating that they may have homology. Besides, the upper part of the Kelulun River is densely populated with industrial parks, with leather processing plants and nitrate plants [48,49], which might be contributed to the higher Cu and Zn concentrations in the southwestern part of the lake. Therefore, factor 3 was associated with industrial sources.

In this work, source apportionment of the ecological risks was assessed by combining PMF and PERI [50], which calculation methods were exhibited in the supplementary material. The source contributions to ecological risk were shown in Figure 7b. Cu-Zn mining and industrial activities were considered to be the the main factor, contributing 66.1%, which exceeded 0.2 times the contribution derived from content. The ecological risk contribution of agricultural sources and natural source were 23.75% and 10.15% respectively, both of which are lower than the contribution values from the content. The change of source contribution was related to the toxic reaction factors of heavy metals [50]. In brief, heavy metals from industrial sources have a higher toxicity coefficient than other sources.

*3.5. Heavy Metals Temporal Variation in the Core Sediments*

The vertical distribution characteristics of lake sediments are considered the records of historical sedimentation of the geographical environment. The vertical changes of element content in rock cores reflect the sedimentary conditions in different historical periods, which can understand the accumulation and superposition history of heavy metals in the study area and reflect the impact of human activities on heavy metals in the study area in different historical stages [49]. In our previous research, the sedimentation rate of the Hulun Lake was calculated as 0.51 cm/a [51]. The sediment deposition rate obtained is similar to that previously reported [41]. It can be seen from Figure 8 that the content of heavy metals changes obviously in stages. Since the Cd concentration is low and the changing trend is not apparent, vertical analysis is not conducted. Heavy metals were detectable in the core sediment, attributed to their constant use and continuous lake input during the study period. The heavy metals concentrations of core sediments at 0–2 cm depth (corresponding to the years 2020–2016) were the lowest, which might be attributed to the remarkable achievements in ecological environment protection and control in the Hulun Lake basin in recent years [51]. This result is in approximate agreement with the results for heavy metals in the surface sediments (Table 1). The maximum peak occured at 6–8 cm, which corresponds to 2008–2004. From 2003 to 2008, Inner Mongolia's GDP growth rate of 17.5% was the highest since the last 20 years and is considered to be the fastest growing economy in China in this time [51]. Heavy metals content in this stage was low and in a stable fluctuation condition between 1978 and 1992, which might respond the background concentration of heavy metals. The

economy was still in its early stages of development, moving slowly, and there were fewer heavy metal emissions due to human activity.

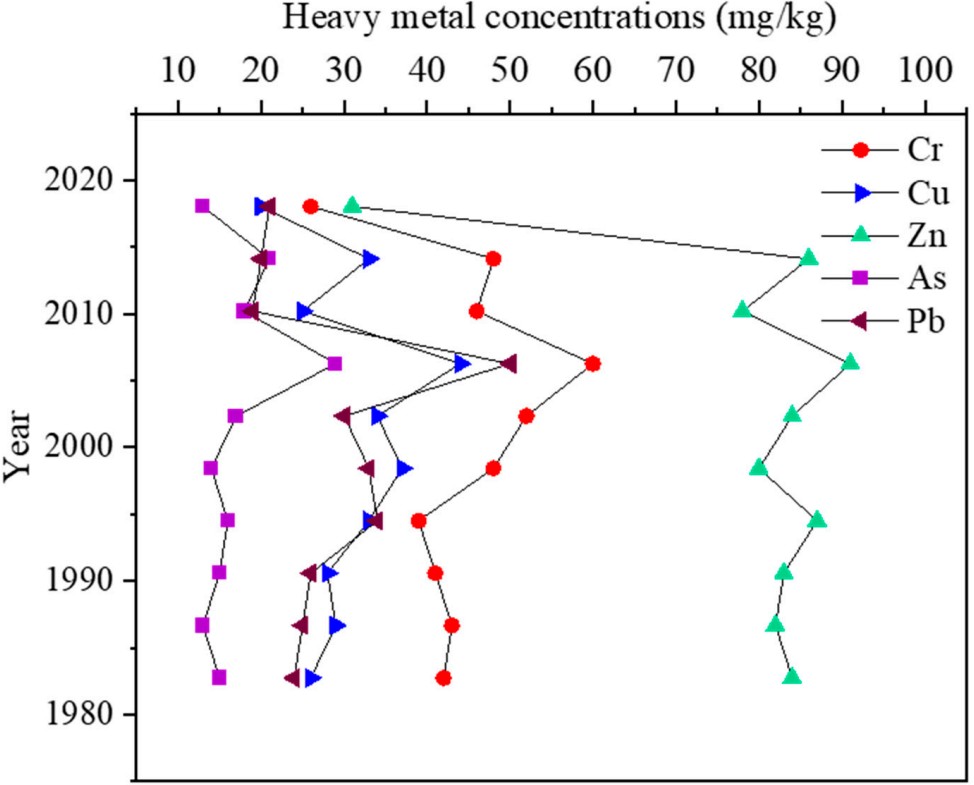

**Figure 8.** Temporal distribution profiles of heavy metals in the core sediments of Hulun Lake.

## 4. Conclusions

Heavy metals in lake sediments threaten aquatic ecological environments and public health. In summary, Cu, Pb, Cr, Zn, Cd, and As were widespread in all surface sediments ranging from 2.96–30.21 mg/kg, 5.39–28.93 mg/kg, 9.20–34.11 mg/kg, 24.50–116.10 mg/kg, ND-0.85 mg/kg and 1.50–14.8 mg/kg, respectively. The average concentration of Cd was below the corresponding background value. The distribution of heavy metals depends on population density and intensity of human activity, and heavy metals were terrestrial pollutants. Surface runoff and suspended particulate matter might be the pathways that carry pollutants from the catchment source to the lake. Three primary sources of heavy metals were identified in sediments: the natural parent material source (relevant indicator is Pb), agricultural sources (relevant indicators are As, Cu and Cd), and mining-related industrial sources (relevant indicators are Cu and Zn), whose contributions to heavy metals' pollution was 17.03%, 26.34% and 56.63%, respectively. The findings of the source apportionment of ecological risks showed that industrial sources were the major ecological risk sources (66.1%), followed by agricultural sources (23.75%) and natural sources (10.15%). The PI and Ieo results indicated that the surface sediment were mainly polluted by Cd, followed by Zn and As. Heavy metal concentrations fluctuate with the year, reflecting a gradual increase in heavy metal use and lake inputs as time passes. The highest peak in heavy metals might be related to the high growth rate of GDP during the same period. In this study, the spatial and temporal distribution of heavy metals in Hulun Lake sediments and the analysis of pollution sources were investigated. The study provides data to support future environmental protection in related fields. The results of the study can provide basic data for the health of the aquatic ecosystem of Hulun Lake, as well as to better protect the fishery resources and environment of Hulun Lake waters and maintain the aquatic environment.

**Supplementary Materials:** The following supporting information can be downloaded at: https://www.mdpi.com/article/10.3390/w15071329/s1, Table S1: The evaluation standard of Geo-accumulation index (Igeo).

**Author Contributions:** T.L.: Writing—original draft, Investigation, Visualization. D.Z.: Methodology, Formal analysis, Data curation. W.Y.: Data curation, Investigation, Funding acquisition. B.W.: Software. L.H.: Visualization. K.L.: Formal analysis. B.-T.Z.: Supervision, Conceptualization, Resources. All authors have read and agreed to the published version of the manuscript.

**Funding:** This work was supported by the National Key R & D Program of China (2021YFC3200101, 2018YFD0900805), Key R & D projects in Hebei Province (21373904D) and Experimental Project of Seasonal River Ecological Flow Technology.

**Data Availability Statement:** Not applicable.

**Conflicts of Interest:** The authors declare that they have no known competing financial interest or personal relationships that could have appeared to influence the work reported in this paper.

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
