# Peer review of "Heavy Metals in Sediments of Hulun Lake in Inner Mongolia: Spatial-Temporal Distributions, Contamination Assessment and Source Apportionment"

_water, doi:10.3390/w15071329_

Round 1

Reviewer 1 Report

In this article, the spatial-temporal distributions, contamination assessments and source apportionment of six heavy metals in sediments from Hulun Lake (Inner Mongolia) have been performed. I would accept this article after minor revisions by taking into account these recommendations.

QUESTIONS AND COMMENTS:

- VERY IMPORTANT For ALL references listed: Check the DOIs, and write it correctly to get direct access: For example, for reference 1 the DOI is: https://doi.org/10.1016/j.marpolbul.2008.07.006

The  current DOIS are not correct.

- In line 93, please specify the brand and model of the sediment sampler used.

- In this way, specify brand and model for the electronic scale, oven, sieve, etc. (lines 99-102). Revise the entire manuscript.

- Line 99 is “each” not “Each”

- What is the particle size of the sediments analysed? (for example 0.125 nm, etc)

- How many replicates do you analyse per sample? Specify the value of n in methodology and tables headers.

- The number of equations are duplicated (see (1) (1), etc…)

- For a better understanding, in line 110, I suggest to name the meaning of CI first and then SI, as appeared in the equation 1, and delete the dots (CI and SI)

- In methodology, I suggest to add a section about the different software (name and versions) used in this manuscript for Figures.

- In line 128: the objective function Q is related to equation 3? If not, describe why equation 4 is also used.

- I am very interested in get access to ref. 43 to see the study about the background levels concentrations of heavy metals in the lake, but I can not find it. Could you please help me with it?

- Where are the LEL and SEL values taken from? Write the reference below table 1, as done for background values.

- For concentration ranges, use “-“ not the symbol used, which means approximately.

- Specify in the title of Table 2 if these values are mean concentrations¿?

- In table 2, column of Cr, please fit the data to cell width.

- I suggest to add the values of the present study to table 2, for a better comparison among them.  

- In figure 3, y-axis, write “PI” instead of “Pi”

Reviewer 2 Report

General, I do not recommend this paper be published at present status. The spatial and temporal distributions, contamination evaluation, and source apportionment of Cu, Zn, As, Pb, Cd, and Cr in the sediments of Hulun Lake were explored in this work. Unfortunately, the paper is just like a report. The discussion of the investigation results lacks depth. More importantly, authors must clearly state the novelty of the results obtained.

* I regret to advise you that the manuscript needs extensive professional language editing to improve the quality of the English. There are many grammatical errors in this manuscript.

* The abstract resumes the starting situation and knowledge of the problem to be approached, the major objectives, the adopted methods and the conclusions. However, the highlights of the study should also be well addressed in this part.

* The introduction is well done in a general way. However, some of the cited bibliographical were too old to response the progress accurately. Meanwhile, the purpose and the main problems needed be addressed in this paper should be well descripted in this part. The innovation/highlight   point of this paper should also be reflected accurately in this part. What are the problems that the research area needs to solve at present?

Line 64-72 These problems lack the necessary research basis. Need to supplement references.

* 2.1. Sample Collection and Physicochemical Analysis: The surface sediment samples were gathered at a depth of 2 cm. However, The sampling depth is not enough to reflect the characteristics of lake bottom sediments. What is the diameter of the sampler? Detailed sampling methods need to be further described. Also, Detailed sample storage methods need to be further described (line 95-96). The sample analysis method needs to be elaborated. What about the analysis quality control strategy and the analysis analytical precision?

Line 146 What does SQG mean?

Line 151-162 What is the significance of such comparison? The sedimentary environments of these lake sediments vary greatly. We should compare lake sediments with river sediments. Especially the estuarine sediments of rivers entering the lake.

Line 177-180 The conclusion here lacks direct evidence.

Fig. 1. There only 12 samples points in the area of 2,339 square 83 kilometers. The reference value of the spatial distribution characteristics of heavy metals in surface sediments is not significant.

Fig. 4 I really don’t understand this figure. Maybe it should be further explained.

Line 243 factor 1 was defined as the natural parent material source. Is there any direct evidence here to support these conclusions.

* Source Identification of Heavy Metals in Surface Sediments by PMF: The analysis of pollution sources should be combined with industrial and agricultural production activities in the study area. Therefore, key industrial and mining enterprises and land use in the study area should be indicated in Figure 1

Reviewer 3 Report

The present work focuses on the caharacterization of heavy metal contamination in lake sediments and potential contamination sources. The overall work is well organized and proper discussion is provided about the obtained results. Accordingly, the work still needs some minor more improvements. As follows the comments:

Introduction: The Introduction should be improved by providing some literature backgorund on strategies aimed at the monitoring and assessment of heavy metals contamination in aquatic environment. Many recent studies are reported in literature. As examples:

https://doi.org/10.1007/s10661-022-10617-4

https://doi.org/10.1016/j.marpolbul.2021.113274

- https://doi.org/10.1007/s13201-018-0815-6 

https://doi.org/10.1016/j.envpol.2020.116212 

 https://doi.org/10.1080/10934529.2017.1397443

Discussion: The overall discussion could be improved by providing a more thorough description about how potentially implement their work in environmental monitoring strategies. How could the present work approach be fitting with the current monitoring practices and, if expected, how could it improve them? Providing this information could be useful to highlight the significance of this work.

Conclusions: "Therefore, this research can offer fundamental knowledge for future environmental preservation actions.", same as the previous comment. You can provide a more detailed explanation of this point maybe adding a new sub-section in the Results and Discussion section about the practical applicability of your work. Moreover, highlight in the Conclusions section what future perspectives should be worth of further investigation on this topic.
